https://doi.org/10.1038/s42003-020-01245-0　　**OPEN**

# Large male proboscis monkeys have larger noses but smaller canines

Ikki Matsuda [1,2,3,4✉], Danica J. Stark[5], Diana A. Ramirez Saldivar[5,6], Augustine Tuuga[6], Senthilvel K. S. S. Nathan[6], Benoit Goossens[5,6,7,8], Carel P. van Schaik[9] & Hiroki Koda [10✉]

The uniquely enlarged noses of male proboscis monkeys are prominent adornments, and a sexually selected male trait. A recent study showed significant correlations among nose, body, and testis sizes and clear associations between nose size and the number of females in a male's harem. However, to date, the analyses of other common male traits, i.e., canines, are lacking. Whereas male nose size had a positive correlation with body size, we unexpectedly found a negative correlation between body and canine sizes. We explain this by an interaction between sexual and natural selection. Larger noses in males may interfere with the use of canines, thereby reducing their effectiveness as weapons. Additionally, longer canines are opposed by natural selection because the larger gape it imposes upon its bearer reduces foraging efficiency, particularly in folivores. This unique case of decoupling of body and canine size reveals that large canines carry an ecological cost.

[1] Chubu University Academy of Emerging Sciences, 1200, Matsumoto-cho, Kasugai-shi, Aichi 487-8501, Japan. [2] Wildlife Research Center of Kyoto University, Kyoto, Japan. [3] Japan Monkey Centre, Inuyama, Japan. [4] Institute for Tropical Biology & Conservation, Universiti Malaysia, Sabah, Malaysia. [5] Danau Girang Field Centre, c/o Sabah Wildlife Department, Sabah, Malaysia. [6] Sabah Wildlife Department, Sabah, Malaysia. [7] Sustainable Places Research Institute, Cardiff University, Cardiff, UK. [8] Organisms and Environment Division, Cardiff School of Biosciences, Cardiff University, Cardiff, UK. [9] Anthropological Institute and Museum, University of Zurich, Zurich, Switzerland. [10] Primate Research Institute, Kyoto University, Inuyama, Aichi 484-8506, Japan. ✉email: ikki-matsuda@isc.chubu.ac.jp; koda.hiroki.7a@kyoto-u.ac.jp

Darwinian evolutionary theory explains many male-specific characteristics as a consequence of sexual selection[1]. Large canines in some mammals, including primates, are a typical example of a weapon used in contests with conspecific male rivals[2]. By contrast, ornaments are not involved in combat and instead act as a partner attraction signal (ultimately determined by female choice). However, some male-specific traits are neither weapons nor ornaments, and serve as a badge of male social status, advertising competitive potential with respect to other males[1,3], particularly as observed for social rank-dependent masculine traits in some primate species[4]. Additionally, such badges of status may be related to animal social recognition[5]; status badges may be beneficial in large/complex societies like primate multilevel societies, in which two or more levels of organization are recognizable. A status badge would therefore provide a useful solution in lieu of individual identification for acquiring an understanding of the relative dominance of conspecifics[6,7].

Proboscis monkeys (*Nasalis larvatus*) are a typical sexually dimorphic primate with male-specific enlarged noses that are prominent adornments, which have been linked with their sexually competitive social system of one-male groups, suggestive of a multilevel social system[8]. Recently, we reported significant positive correlations among nose, body, and testis size as well as a clear association between nose size and both nasalized acoustic signals and the number of females in harem groups; this supports the hypothesis that larger noses audiovisually advertise a male's fighting ability with conspecifics[9], and thus serve as status badges rather than ornaments.

To date, an analysis of another common sexually selected male trait, i.e., canine size, is lacking. Sexually dimorphic male traits (e.g., body mass, nose, testis, and canines) develop after sexual maturation, primarily in response to endocrine changes in the levels of androgens such as testosterone[10–12]. Comparisons of these male traits would enable us to study the relationships among weapons, badges, ornaments, and testes. In this study, we compared canine size among previously investigated wild proboscis monkeys[9]. Notably, proboscis monkey harem groups (and bachelor groups) aggregate on a regular basis at their sleeping sites, and female dispersal between the groups is typical[13] in their multilevel society[8], creating the potential for both female choice and male–male competition for females. Therefore, male canines are expected to serve as important weapons for mate competition or defense, especially because canine size generally reflects the strength of intra-sexual selection more directly than body size in interspecific comparisons[2,14]. Together with the evaluations by measuring the body traits, we attempted to test whether the development of canine size is replicated by a mathematical simulation with minimal assumptions, and search for the plausible hypothesis to explain the multiple sets of the biological properties of the proboscis monkeys that are often exposed to the natural and sexual selections.

## Results

**Relationships among physical traits**. The best-fitting model included both male nose and canine size, although the fit of the model including only canine size was almost as good (i.e., ΔAICc < 2.0: Table 1). Whereas nose size was positively correlated with body mass, the model with canine size was unexpectedly negative (Fig. 1). As the values for the young/old adult males were within the range of the ones for the "averaged" adult males (Fig. 1), we, therefore, believe that age/wear of each individual male would not contribute greatly to the relationship between body size and canine size. However, due to the limited sample size, further analysis using a larger data set would be necessary to reconfirm these results in a future study. Conversely, for females, the best-fitting model only included nose size, which was positively associated with body mass. The second-best model was the null model (Table 1). The difference in the regression slopes and intercepts for body size vs. nose size between the sexes was not significant (slope: $t = -1.41$, $P = 0.17$; $t = 0.28$; intercept: $P = 0.78$). However, the difference in the regression intercept for body size vs. canine size was significant ($t = 8.28$, $P < 0.0001$), although the slopes were not different ($t = -0.33$, $P = 0.74$). Previous results illustrated that secondary sexual proboscis monkey male traits reflect both pre- and post-copulatory competition; specifically, larger noses serve as both "female attraction" and "badge of status," and simultaneously, larger testes reflect sperm competition[9]. Body size also contributes to both male–male competition and female attraction.

**Numerical simulation**. Figure 2 shows the simulated results of developmental growth curves for harem alpha status males, non-alpha status males, and females with various cost and termination time parameters. Our simulations revealed that higher costs caused by canine development negatively influenced body mass growth in males. Male body mass also varied depending on the termination time of canines, i.e., canine size, whereas female body mass was relatively independent of canine size. In the case of no or weak costs attributable to canine development, body mass for both sexes was independent of the canine growth termination time, but under the assumption of a large cost for canine development, canine size had a clear negative influence on body mass, particularly in males (Fig. 3).

## Discussion

The results presented in this study can be interpreted in two ways. The minimal interpretation is that canine size does not contribute to the fighting ability of harem-holding males, unlike the findings in other primate species forming harem societies[2]. Battles featuring fatal physical attacks using canines are commonly observed in typical harem species such as hanuman langurs (*Semnopithecus entellus*)[15] or howler monkeys (*Alouatta seniculus*)[16]. Nonetheless, the benefit of a larger body size in proboscis monkey males is clear. Indeed, branch shaking is commonly used to keep groups apart[8], and in chases without physical contact, males repeatedly exhibit such displays accompanied by leaps between trees[13,17]. Additionally, among proboscis monkey males, nose size, which is highly correlated with body size, determines the formant frequencies of loud vocalizations, which may decide

---

**Table 1 Summary of model selection.**

| Intercept | Nose size | Canine size | d.f. | Log likelihood | AICc | Δ−AICc | AICc weight |
|---|---|---|---|---|---|---|---|
| Male | | | | | | | |
| 42.3 | 0.11 | −0.01 | 4 | 25.0 | −39.0 | 0 | 0.41 |
| 33.0 | | −0.01 | 3 | 23.2 | −38.7 | 0.34 | 0.34 |
| Female | | | | | | | |
| 1.39 | 0.27 | | 3 | 12.3 | −14.6 | 0 | 0.77 |

Linear models were used to investigate whether body mass was related to other body traits such as nose and canine size (only the models with ΔAICc ≤2 are shown).

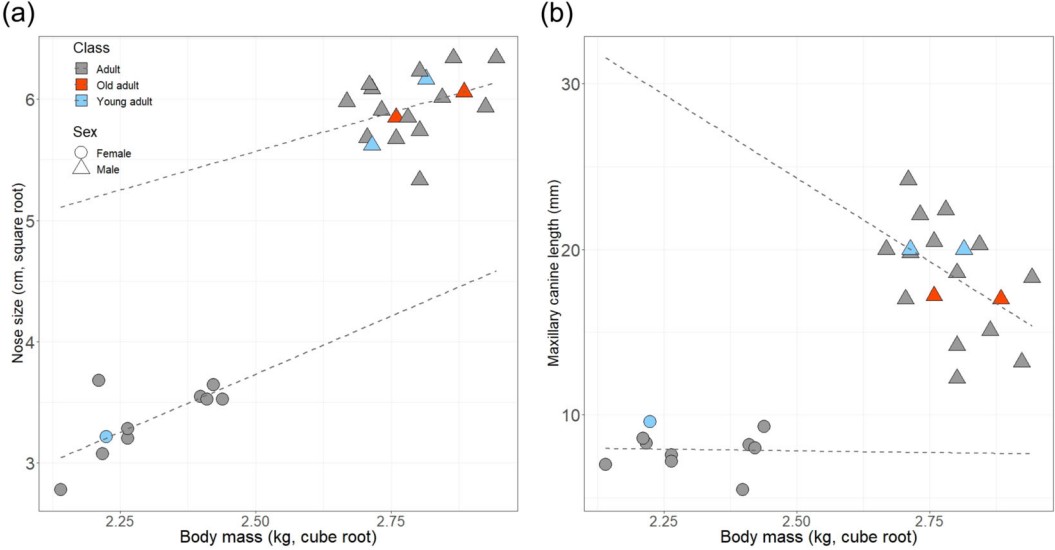

**Fig. 1 Relationships of nose size and canine size and with body mass.** Nose size was positively and significantly correlated with body mass for both sexes **a**, though a significant negative correlation between body and canine size was found only in males but not in females **b**. Noted that sex and estimated age/class category is indicated in different shape and colour symbol, respectively.

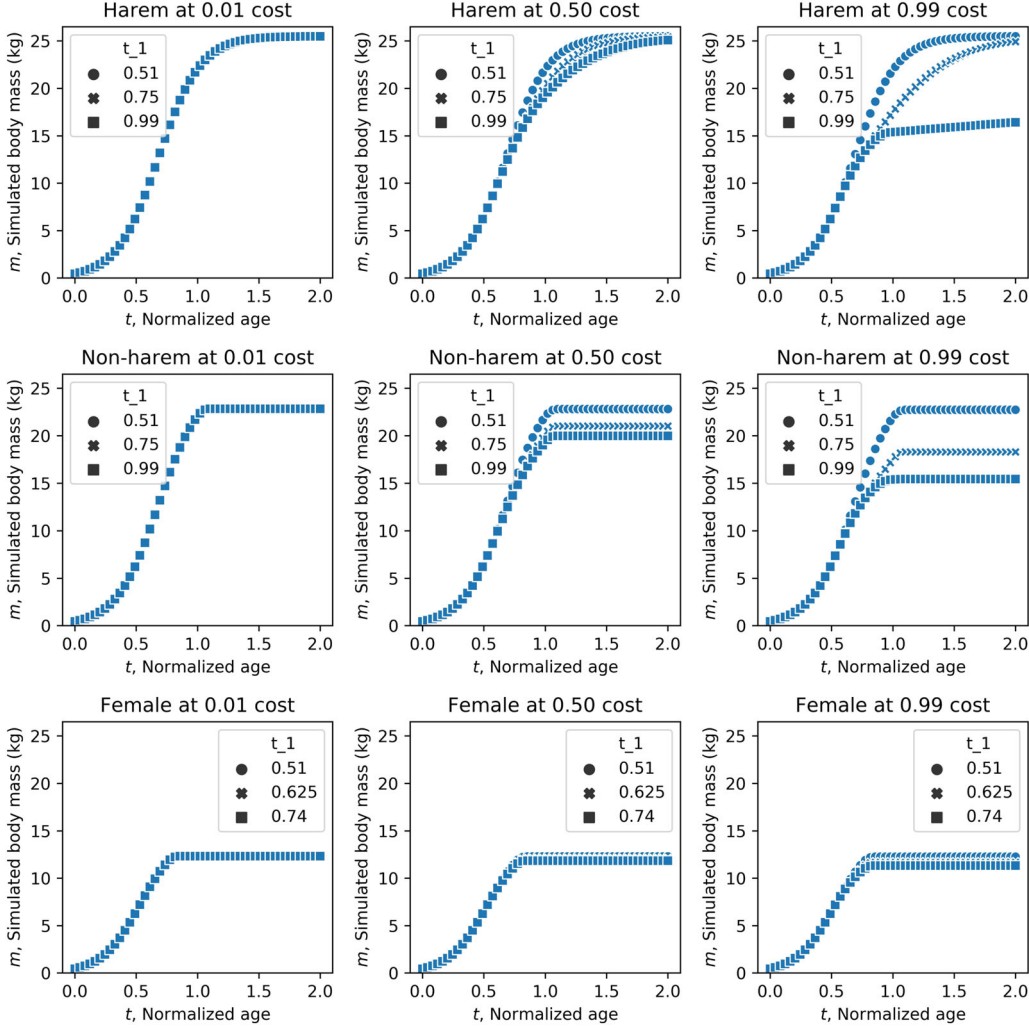

**Fig. 2 Developmental growth curves for harem males, non-harem males, and females.** The cost parameters were set to 0.01 (weak), 0.5 (moderate), or 0.99 (high). The termination of canine development (represented as "t_1" in figures) was set to 0.51 (immediate termination), 0.75 (middle), or 0.99 (maximum canine growth) for males and 0.51 (immediate termination), 0.625 (middle), or 0.74 (maximum canine growth) for females.

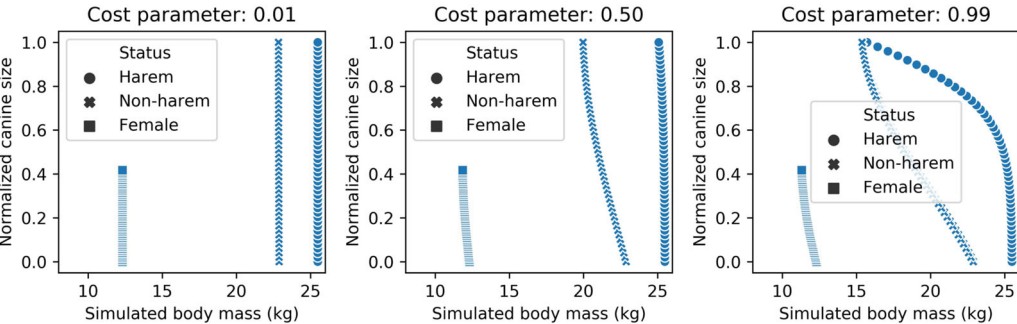

**Fig. 3 Relationships of normalized canine size with body mass at the final developmental stages for simulated harem status males, non-harem status males, and females with various cost assumptions.** Left: weak cost ($c = 0.01$); middle: moderate cost ($c = 0.5$); right: high cost ($c = 0.99$).

contests without any direct physical competition, or at least direct biting[9,13,18]. Thus, a larger body, which contributes to male athleticism, would allow more effective displays of fighting ability, whereas nose size would be a more suitable proxy to represent fighting ability because it is easier to assess via vocalization as opposed to visual examination. Consequently, males might invest their physiological resources to develop nose, body, and testis size more than canine weapons, in line with the trade-off phenomenon recently reported in other harem primates, e.g., Dunn et al.[19].

Contrarily, whereas females do not compete through sound as do males, they unexpectedly exhibited the same nose size/body size correlation, suggesting that female nose size is linked to the effect on males; positive selection on larger male nose size may be driving larger noses in females. Female noses are probably sufficiently small to avoid interfering with their activity (e.g., feeding, moving, social behaviors, etc.), and thus, selection toward sex-differentiated expression does not occur. In other respects, it has been found that secondary sexual characteristics, i.e., ornaments or weapons, have a positive allometric slope concerning body size in a number of animal species[20,21], a positive correlation between nose size and body size in either sex would just be a natural phenomenon and requires no complex such explanation.

However, the significantly negative correlation between male canine and body size supports an alternative interpretation; namely, it suggests an ecological cost of large canines, as suggested for the taxonomic variation in canine size observed across primate lineages[22,23]. The prominent nose may interfere with effective canine bites, thereby reducing their efficacy as weapons. Moreover, given their reduced effectiveness, natural selection may actually favor smaller canines because larger canines increase gape, which reduces bite force and thus mastication efficiency[22]. This foraging cost may be especially severe in large-sized folivores, such as proboscis monkeys and gorillas[23]. Thus, a negative correlation might evolve in the rate of post-adolescent growth of the size of the body frame and that of canines. We therefore predict that in proboscis monkeys, body-size dependent displays are more effective at greater distances than the facial displays commonly observed in most other primates[24].

Contrary to the significant negative slope for the male canine size–body size correlation, we only found a weak negative correlation for the female slope. Considering lower incidence of agonistic interaction, either using canines and leaping between trees, with weak or absence of linearity in hierarchy among females within groups[13,25], such canine size–body size correlation may be less prominent in females than that in males. Although the difference of the slopes between the sexes was not significant, possibly because of the limited sample size, the

significant sex difference in the intercept for body size vs. canine size suggests that canine size has a different relationship with body mass between males and females. The difference in the sex-differentiated pattern between noses and canines would confirm the hypothesis that large canines interfere with effective biting in males, making them less effective as weapons and thus leading to negative selection on canine size in the largest males.

Concerning the mechanism underlying the trade-off among body mass, nose size, and canine size, we predict that canine growth would actually exhibit a negative correlation possibly with testosterone levels, opposite to the common and ancestral situation[19]. To test a part of this "costly canine hypothesis", we conducted preliminary mathematical simulations considering body mass development with a growth constrained by canine size. Echoing our empirical findings, the negative relationship between body and canine size was observed only in males when we assumed a large cost of canine development (Fig. 3); noting that the model did not consider the possible basal factors such as testosterone level which might fundamentally determine the maturation patterns of the sexual traits, due to no available data for the initial parameter proposals. Our study, however, raises interesting questions regarding the developmental mechanism that produces a negative correlation among adult body, nose size, and canine size. Future detailed studies focusing on the behavior and development system of proboscis monkeys are therefore required.

## Methods
**Ethics statement.** Permission to capture and handle proboscis monkeys was granted by the Sabah Biodiversity Centre (permit JKM/MBS.1000-2/2 JLD.3 (73)) and was carried out in accordance with the current laws of Malaysia, Sabah Wildlife Department's Standard Operation Procedures on Animal Capture, Anaesthesia and Welfare, the Weatherall report[26], and the guidelines for non-human primates as described by Unwin et al.[27]. Once an risk assessment was conducted on the target individual and its surrounding sleeping site area, darting was performed by veterinarians experienced in the capture and anesthesia of wildlife using Zoletil 100 (Tiletamine + Zolazepam; 6 ± 10 mg/kg), Anaesthesia and the vital signs were monitored throughout the procedure[28], and once the procedure was complete, each animal was given a prophylactic dose of Alamycine LA (20 mg/kg) and Ivermectine (0.2 mg/kg) as a preventative measure to assist in the post-anesthesia recovery.

**Data collection.** Between July 2011 and December 2016, we captured 28 free-ranging adult proboscis monkeys in the Lower Kinabatangan Floodplain, Sabah, Borneo, Malaysia (5°18′N to 5°42′N and 117°54′E to 18°33′E). To reduce the impact of capturing on the animal's social system, we captured all study subjects during the night[28]. While animals were anaesthetized, we performed in situ measurements for their body parts using a scale (body mass) and caliper (nose size and canine length). In this study, we additionally included data for maxillary canine length to those on body mass and nose size (length × width) for the same 18 males obtained by Koda et al.[9] as well as data for 10 females' canine length, body mass, and nose size. Note that canine length refers to the apex to base measurement (the height of the crown), and the maxillary canine length was measured because of

its larger size and greater importance in behavioral displays and weaponry[29]. All study subjects were adults, i.e., 18 harem-holding males and 10 females including one pregnant and two lactating individuals (see Supplementary Table 1 for further detailed study subject information). The veterinarian in the sampling team attempted in situ age/class estimation for the study subjects based on the three categories of wear level of their molars, i.e., low: young adult; medium: adult; high: old adult, although the estimation was rather rough due the difficult in situ condition in the forest at night, e.g., limited tools, light, and time the animal remained anaesthetized.

**Data analysis**. We used a linear regression model to establish whether body mass (response variable) was related to other physical properties such as nose and canine size (explanatory variables). To obtain a linear dimension comparable to the canine dimension, body mass and nose size were cube root- and square root-transformed, respectively, to generate response and explanatory variables. The variance inflation factors were 1.00 for both nose size and body mass, indicating that collinearity among independent factors would not affect the results[30]. We examined a set of models with all possible combinations of the explanatory variables and ranked them according to the corrected version of the Akaike information criterion (AIC) for small sample sizes, called AICc[31]. Following guidelines published for wildlife research, we selected models with $\Delta$AICc $\leq 2$, where $\Delta$AICc = AICc−minimum AICc within the candidate model set[31]. Analysis of covariance served to compare the slopes and intercepts of regression lines between the sexes. We performed these analyses using R ver. 3.3.2 (ref. [32]).

**Mathematical model**. The aim of the simulations was to examine our hypothesis regarding the positive/negative correlations between body mass and canine size. Earlier work distinguished the subadult class from fully adult males based on nasal maturation as well as a fully developed body size[9]. In other words, two developmental stages likely exist among sexually mature males. Therefore, before the acquisition of harem status, subadult males reach a limit in body mass, which cannot increase without a status change. By contrast, in females, the development of body mass and canines is basically similar to that of males before the acquisition of harem status. In this study, we therefore assumed a two-stage model of male development, i.e., a first stage with primary development of body and canine size for both males and females and a second stage that only applies to males and only after the acquisition of harem status.

*Primary development of body and canine size*. Body mass $m$ develops under the assumption that the growth rate depends on the body mass at the time, until reaching its developmental limit $K$ within a fixed time $T$, i.e.,

$$\frac{\mathrm{d}m}{\mathrm{d}t} = am\left(1 - \frac{m}{K}\right) \quad \forall t \in [0, T], \tag{1}$$

where $a$ = const is the shape factor determining the sigmoidal curve of development. This is based on a previously proposed basic model for body mass development[20,33] but with our modification to simplify the constraint of $K$. By contrast, the canines develop in the later stage of body mass development, and thus, we supposed that larger canines would reduce feeding efficiency, resulting in a growth rate reduction for body mass as follows:

$$\frac{\mathrm{d}m}{\mathrm{d}t} = (1 - cz)am\left(1 - \frac{m}{K}\right), \tag{2}$$

where $cz$ is the reducing factor determined by the cost factor $c \in [0, 1)$ and the size of canines $z$ and $cs < 1$ always holds. For model simplification, the canines start to develop at the time $t_0$ and linearly develop until $t_1 \leq T$ as follows:

$$z = \alpha(t - t_0) \quad (t_0 \leq t \leq t_1), \tag{3}$$

where $\alpha > 0$ is the growth rate of canines. Consequently, the differential equation of body mass in $t \in [0, T]$, $T > t_1$ is

$$\frac{\mathrm{d}m}{\mathrm{d}t} = \begin{cases} am\left(1 - \frac{m}{K}\right) & (0 \leq t \leq t_0) \\ \{1 - c\alpha(t - t_0)\}am\left(1 - \frac{m}{K}\right) & (t_0 \leq t \leq t_1) \\ (1 - cz_1)am\left(1 - \frac{m}{K}\right) & (t_1 \leq t \leq T), \end{cases} \tag{4}$$

where $cz_1 := cz_{t=t_1} = c\alpha(t_1 - t_0)$, because proboscis monkeys likely increase their body mass until $t = T$, with the constant of the reducing factor, after terminating canine growth ($t = t_1$). Note that $0 \leq z \leq \alpha(T - t_0)$ holds; therefore,

$$\alpha \leq \frac{z}{T - t_0}, \tag{5}$$

also holds.

In the case of $t < t_0$, i.e., before the onset of canine eruption, the equation is a simple logistic equation; therefore, the solution is

$$m(t) = \frac{K}{1 + A_0 \mathrm{e}^{-at}}, \tag{6}$$

where $A_0$ is the constant determined by the initial values of $m(0) = m_0 > 0$ as

follows:

$$A_0 = \frac{K}{m_0} - 1. \tag{7}$$

Once the canines begin to develop, i.e., in the case of $t_0 \leq t \leq t_1$, the differential equations are solved as follows:

$$\begin{aligned} \frac{\mathrm{d}m}{\mathrm{d}t} &= \{1 - c\alpha(t - t_0)\}am\frac{K-m}{K} \\ \int \frac{K}{m(K-m)}\mathrm{d}m &= a\int\{1 - c\alpha(t - t_0)\}\mathrm{d}t \\ \int\{\frac{1}{m} + \frac{1}{(K-m)}\}\mathrm{d}m &= a\int\{1 - c\alpha(t - t_0)\}\mathrm{d}t \\ -\ln\left|\frac{K-m}{m}\right| &= -\frac{1}{2}ac\alpha t^2 + a(1 + c\alpha t_0)t + C \\ m(t) &= \frac{K}{1 + A_1 \mathrm{e}^{\frac{1}{2}ac\alpha t^2 - a(1 + c\alpha t_0)t}}. \end{aligned} \tag{8}$$

Then, after terminating canine development ($t = t_1$), we found that the solution is simply

$$m(t) = \frac{K}{1 + A_2 \mathrm{e}^{-a(1 - cz_1)t}}. \tag{9}$$

Note that the following equations must be satisfied:

$$\begin{aligned} m(t_0) &= \frac{K}{1 + A_0 \mathrm{e}^{-at_0}} = \frac{K}{1 + A_1 \mathrm{e}^{\frac{1}{2}ac\alpha t_0^2 - a(1 + c\alpha t_0)t_0}} \\ m(t_1) &= \frac{K}{1 + A_1 \mathrm{e}^{\frac{1}{2}ac\alpha t_1^2 - a(1 + c\alpha t_0)t_1}} = \frac{K}{1 + A_2 \mathrm{e}^{-a(1 - cz_1)t_1}}. \end{aligned} \tag{10}$$

Consequently, the basic models for body mass are

$$m(t, t_0, t_1, K, a, c, \alpha, A_0, T) = \begin{cases} \frac{K}{1 + A_0 \mathrm{e}^{-at}} & (0 \leq t \leq t_0) \\ \frac{K}{1 + A_0 \mathrm{e}^{\frac{1}{2}ac\alpha t_0^2 + \frac{1}{2}ac\alpha t^2 - at(1 + c\alpha t_0)}} & (t_0 \leq t \leq t_1) \\ \frac{K}{1 + A_0 \mathrm{e}^{\frac{1}{2}ac\alpha t_0^2 - \frac{1}{2}ac\alpha t_1^2 - at + ac\alpha t_1 t - ac\alpha t_0 t}} & (t_1 \leq t \leq T), \end{cases} \tag{11}$$

and those for canine size are

$$z(t, t_0, t_1, \alpha, T) = \begin{cases} 0 & (0 \leq t \leq t_0) \\ \alpha(t - t_0) & (t_0 \leq t \leq t_1) \\ \alpha(t_1 - t_0) & (t_1 \leq t \leq T). \end{cases} \tag{12}$$

*Rank-dependent secondary development of body and nose size*. Our previous study suggested that males who acquire harem alpha status develop their noses as a badge of status in coordination with body mass[9]. In this study, we supposed that alpha status males can dominate both copulation opportunities and foraging resources because harem groups with larger males better defend resources from bachelor groups generally consisting of smaller males[34]. Therefore, such alpha status males would possibly continue their body mass growth after completing their primary developmental process, whereas males who fail to acquire harem status terminate body mass growth. Then, the differential equations of the body mass in the secondary development for harem males are equivalent with Eq. (4), i.e.,

$$\frac{\mathrm{d}m}{\mathrm{d}t} = (1 - cz_1)am\left(1 - \frac{m}{K}\right) \quad (t \geq T), \tag{13}$$

whereas those for non-harem males are

$$\frac{\mathrm{d}m}{\mathrm{d}t} = (1 - cz_1)\{a - \frac{t - T}{t_2 - T}a\}m\left(1 - \frac{m}{K}\right) \quad (t \geq T), \tag{14}$$

where $t_2$ is the termination time of the body mass growth. This equation is solved in the same manner as Eq. (8), namely

$$\begin{aligned} m(t, t_0, t_1, K, a, c, \alpha, A_0, T, t_2) &= \frac{K}{1 + A_3 \mathrm{e}^{\frac{1}{2}\beta t^2 - \beta t_2 t}} \\ A_3 &= A_0 \mathrm{e}^{\frac{ac\alpha(t_0^2 - t_1^2)}{2} - aT + ac\alpha T(t_1 - t_0) - \beta T(\frac{1}{2}T^2 - t_2 T)}, \end{aligned} \tag{15}$$

where $\beta := \frac{a(1 - c\alpha t_1 + c\alpha t_0)}{t_2 - T}$. Based on our findings of allometric development of ornaments and body size in free-ranging specimens[20,33], we supposed that nose size is primary determined by body mass.

**Numerical simulations**
*Overview of aims*. We conducted the numerical simulations with the aim to clarify the mechanism by which the termination time of canine development $t_1$ determines the final results of body mass and canine size. We simulated the growth pattern based on several deterministic parameters, considering the developmental evidence for proboscis monkeys. Additionally, we tried to apply the model for both males and females within the same developmental frameworks, considering the sex differences of the developmental parameters as follows.

*Parameter proposals*. First, the time parameter was normalized at the male's primary maturation time $T$, i.e., $T := 1$. Therefore, our time parameters $t_0$, $t_1$, and $t_2$, are the time relative to the male maturation time $T$. By definition, $0 \leq t_0 \leq t_1 \leq T = 1 \leq t_2$ holds. Additionally, the maximum canine size $z$ was also normalized to 1 for the case in which the monkey maximally develops its canines during the periods between $t_0$ and $T = 1$. Following Eq. (5), $\alpha = \frac{1}{1 - t_0}$.

Next, we supposed that the primary maturation age of males ($t = T = 1$) at which males reach sexual maturity at approximately 8 years old[35], although male nose enlargement does not typically start until that time[36]. In *Cercopithecus*[37] and

proboscis monkeys[38], canine development starts at approximately 4 years old and continues until 8 years old. The onset time of canine development was therefore set at 4 years old in our model, or half the subadult maturation time, i.e., $t_0 = \frac{1}{2}$. By contrast, females mature earlier than males. We supposed that the female maturation age was 6 years[35]; i.e., we used $T = \frac{6}{8}$ in the model for female development. The time of onset of canine growth is likely similar in both sexes, but females likely terminate canine growth earlier, corresponding to their earlier physical maturation. Therefore, we supposed that female canines develop from approximately 4 years old until 6 years old, representing a shorter than observed in males (i.e., $t_0 = \frac{1}{2}$ for females).

Records for body mass were used for the parameters in the simulations. Proboscis monkey neonatal body mass was reported as 0.45 kg[38], and the maximum body mass among our specimens was 25.5 kg for males and 14.5 kg for females. Therefore, we set $m(0) = 0.45$, $K_{male} = 25.5$, and $K_{female} = 14.5$. Additionally, we used the body mass record at approximately 4 years old, or the onset time of canine eruption, which was approximately 6.5 kg ($N = 3$, records in Japan Monkey Centre). Therefore, we used $m(\frac{1}{2}) = 6.5$ in the simulations for both males and females. Finally, we assumed that body mass development would terminate immediately after the maturation time $T$ was reached; therefore, we used $t_2 = T + 0.1 = 1.1$ for males and $t_2 = T + 0.1 = 0.85$ for females.

Based on the aforementioned parameters, we simulated body mass at the time at which monkeys fully develop. For this purpose, we attempted to simulate/evaluate $m$ (2), or body mass at approximately 16 years old, using the various combinations of cost parameters and canine termination times, i.e., $c \in (0, 1)$ and $t_1 \in (0.5, T)$.

**Reporting summary**. Further information on research design is available in the Nature Research Reporting Summary linked to this article.

## Data availability
Data in support of the findings of this study are available from the corresponding authors by reasonable request.

## Code availability
Mathematical models and proposed parameters are also described in the code repositories (https://github.com/hkoda/nasalis_canine_bodymass_model).

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

## Acknowledgements

We thank the Sabah Biodiversity Centre and the Sabah Wildlife Department for granting permission to carry out this research, our research assistants for support in the field, and the Japan Monkey Centre (Dr. N. Kimura) for providing the information used in our mathematical simulations. We appreciate the profound comments and suggestions from the members of "Tsuda CREST", especially I. Tsuda to support verifying the mathematical simulations. This study was partly performed under the Cooperative Research Program at KUPRI (2020-B-17) and was partially funded by Japan Science and Technology Agency Core Research for Evolutional Science and Technology 17941861 (#JPMJCR17A4), Ministry of Education, Culture, Sports, Science and Technology Grant-in-aid for Scientific Research on Innovative Areas #4903 (Evolinguistics), 17H06380, Japan Society for the Promotion of Science KAKENHI (#19KK0191 to IM), and Yayasan Sime Darby (to B.G.). This paper acknowledges the memory of Dr. Diana Angeles Ramirez Saldivar.

## Author contributions

I.M., B.G., C.P.v.S., and H.K. conceptualized the initial idea; I.M., D.J.S., D.A.R.S., and S.K.S.S.N. performed the wild morphology sampling, A.T. and B.G. arranged the sampling in the wild; I.M. and H.K. performed and interpreted the statistical analyses and mathematical simulation; I.M., C.P.v.S., and H.K. drafted the manuscript. All authors contributed to the final version of the manuscript.

## Competing interests

The authors declare no competing interests.
