## [Peer Review File · Communications Biology]

Reviewers' comments:

Reviewer #2 (Remarks to the Author):

This manuscript provides interesting new data on the association between canine length, nose length, and body size in proboscis monkeys. The authors find a surprising negative correlation between body size and canine length in adult males and explain this by suggesting possible ecological costs for having long canines. This work will be of interest to others studying trade-offs in signaling in animals. For the most part the manuscript is well-written and the data are analyzed appropriately. The inclusion of simulations on the effect of costly canine development helps bolster the conclusions. I have just a few comments to improve the clarity and flow.

The Introduction could be improved by citing the two papers by Bergman and Sheehan (2013 and 2016) on quality signaling and individual recognition in primates and discussing the evolution of badges of status in light of the potential multi-level society in proboscis monkeys. Also, Rohwer's (1982) important paper on badges of status should probably be cited (references provided below).

The sample size of males that was actually included in the study needs to be clarified. Lines 76-82 are not clear. Did Koda et al. study the same males as Stark et al.? How many females were included? I think a table detailing the study subjects would be helpful here.

In the Results/Discussion, the paragraph on lines 138 to 142 needs further fleshing out and explanation. On lines 138-140, what is meant by the idea that "female nose size is linked to the effect on males"? Isn't it just that larger individuals have larger body parts and this includes noses? Or are the authors suggesting that positive selection on male nose size is driving larger noses in females? I'm not sure that these two possibilities can be teased out with this data set. It may be best to discuss both. On lines 140-142, what "activity" is being referred to? Canine activity or feeding etc.? This is not clear.

Finally, the passive voice is used often throughout the manuscript. It would benefit from a switch to the active voice.

Minor comments:

L 22-25 – The first sentence is very long. Breaking it up would make it clearer.

L 28 – Add "A" before "similar".

L 33 – I don't think "However" belongs here, since you are presenting an additional reason why longer canines are disadvantageous. "In addition," or something similar would be better.

L 48 – This does not seem right, "ornaments independently evolve for a function in combat." Do you mean "ornaments independently evolve FROM a function in combat and instead act as a partner attraction signal"?

L 83 – Start a new paragraph with the line "A linear model".

L 111 – Since it is likely body mass driving larger noses for females, I think the wording here would be better as, "was positively associated with body mass".

L 134 – Change "vocalization opposed" to "vocalizations as opposed".

I hope these comments are helpful in revising your work.

Julie Teichroeb

References

Bergman, T. J., & Sheehan, M. J. (2013). Social knowledge and signals in primates. *American journal of primatology*, 75(7), 683-694.

Rohwer, S. (1982). The evolution of reliable and unreliable badges of fighting ability. *American Zoologist*, 22(3), 531-546.

Sheehan, M. J., & Bergman, T. J. (2016). Is there an evolutionary trade-off between quality signaling and social recognition?. *Behavioral ecology*, 27(1), 2-13.

Reviewer #3 (Remarks to the Author):

This manuscript reported a negative relationship between canine size and body size in morphologically unique proboscis monkeys. A negative relationship is unexpected from the sexual selection theory but can be understood based on a unique characteristic of proboscis monkeys.

Morphological data in wild proboscis monkeys is valuable. The manuscript is very simple and easy to understand. But I have several questions and comments.

First, I do not think that a section of simulation is necessary. It adds little to the results.

Second, it would help if authors explain details of proximate aspects (mechanism) of canine development. In 62-64, the authors mention that canines develop in response to testosterone. But no citation on these statements. Is this true?

L76-77: 18 free-ranging adult male proboscis monkeys. Are those males harem-holders? Or a mixture of harem-holders and bachelors? This information should be important because authors assume that differences between harem-holders and others are important.

L83-87: Although what is written is correct, I appreciate it if authors could explain which one is a response variable and what are explanatory variables more clearly.

L129-132. Although I am not a native English speaker, I prefer not to use two which in a single sentence.

Table 1. please use the same number of decimal digits for all values.

Reviewer #4 (Remarks to the Author):

This is an interesting paper on the relationship between canine tooth size, body size, and nose size in proboscis monkeys. The authors have some valuable data, and some interesting observations that might have some interesting implications for the evolution of proboscis monkey noses. However, the paper suffers from an inadequate explanation of the materials and methods, inadequate presentation of the results, and failure to consider alternative explanations for the observed results.

The introduction is mostly fine, but there are a few confusing statements. For example, the authors state "By contrast, ornaments independently evolve for a function in combat, reflecting their function as a partner attraction signal (ultimately determined by female choice)." This doesn't make sense. There are few more rough spots in the introduction. I think that the authors need to carefully review exactly what they are saying and make sure that the introduction is both concise and clear.

Methods. The data are referred to another paper, but the reader should not be forced to chase the data down. What exactly is nose length and breadth? How was body mass measured? Were pregnant female included? Did you note lactation? When measuring canines, did you account for

wear? Or did you need to? Canine length can refer to the apex to base measurement (the height of the crown), which has been argued to be the functional aspect of the canine, or the mesiodistal length at the base of the tooth. This makes a huge difference, because if you are measuring canine tooth height, wear will be expected to shorten the tooth with age, and if size increases with age, then perhaps your results are an artifact of age-size and tooth wear. On the other hand, if you are measuring the occlusal dimension of the tooth, then you might run into problems because this aspect of the tooth is not as strongly correlated with behavioral measures as height.

I am not an expert on the mathematical modelling, and so cannot really comment on it. Models are great, but better explanation is needed. After all, the test of the hypothesis should be based on the data. What bothers me here is that I have some fair experience in stats, and when I read a paper I should know exactly what you are doing, and I cannot see that here. How is the termination of canine tooth development determined? These are wild data, and I assume that sample sizes (which I can't determine from this study – all I know is that data for 18 males and an unknown number of females was added to some data from another study that I don't have) are small. In the figure I count 10 females and 18 males, which for the males matches the stated number gathered, but if these were added to other data then something is wrong.

Reporting of the results needs to be much more systematic. The authors have done regressions, ANCOVA, and something else, but I can't be quite sure because the results are not actually here. What I do have is some graphs, and the graphs are suggesting several things. First, nose size and body size are tightly correlated in females, an less so in males, and it looks like a standard allometric phenomenon with an offset between the sexes (the ANCOVA should show no significant interaction and the slope between males and females does not significantly differ). The canines show no relation between female canine size and body size, and the males show a negative relationship, though it looks like the correlation is going to be weak. These are interesting results, but I am not convinced that the authors have considered alternative explanations. First, if male canine size is impacted by wear, but size increases with age (see for example drills), then we should expect canine size to have a negative correlation with body size not through selection, but through tooth wear. Second, male nose size increases with size. If size increases with age, then the model proposed by the authors is not supported, as the results reflect an interaction between age, size, and tooth wear. Alternatively, if male canine tooth eruption is completed before final adult body mass is achieved, then we should expect a loose, negative correlation between tooth size and body size. Again, this would mean that the model of the authors is not supported. I really can't say much more about the results and discussion unless this basic issue is addressed. The authors need to address this issue head on by including age in their estimates. Without that, not much can be said. The models might work under the assumptions that authors input, but I am not sure that the modelling accounts for these two basic features. If it does, the authors need to explain their model better, and it needs to be validated against actual data ("ground tested").

COMMSBIO-19-1713
Reply to Reviewers

Dear Dr. Katie Davis,

Thank you for giving us the opportunity to revise our manuscript and respond to the Reviewers' comments.

Please find attached the revised manuscript with all revised text in **grey shading**. Our response to specific reviewer comments are given at the bottom of this letter.

Generally, all reviewers' comments were really constructive and made our manuscript much better, but please noted that the comments of reviewers #2 and #3 were easier to answer than those of reviewer #4. The comments from Reviewer #4, who appears to be a specialist in primate canines, were particularly helpful to improve our manuscript, especially the comments about the age/class and wear effects on our results. As answered below in detail, we did our best to improve our analysis based on the comments by all reviewers, but we would like to reiterate that our observed data was collected from free-ranging endangered animals, and not from fossils or skeletal measurements. The data for these living animals are very rare and valuable, which means that re-measurements are not possible. We hope that Reviewer #4 and the editor will consider this point. Furthermore, we are concerned that Reviewer #4 might have misunderstood our mathematical simulation– and was confused between the statistical modeling (using the observed data set) and our mathematical simulation based on our hypothesis. As explained below, the aim of our simulation was to evaluate our hypothesis “costly canine hypothesis” proposed by our observed data set (regression data analyses), with the minimal assumptions of the model parameters. Therefore, our simulation is different from the concept of normal statistical modeling. Lastly, we believe that our simulation is indeed a worthy contribution to proposing a challenging hypothesis in biology, though the detailed explanation including many formulas may not necessarily fit your journal, as readers may range from backgrounds as theoretical biologist to more wide ranging biological fields, whereby the detailed descriptions with many formulas may interrupt the smooth flow and reading of our text/findings. For this reason, we believe that the detailed explanation for our simulation part is best kept as supplementary information, and not in the main text, even though Reviewer #4 had requested us to include more detailed descriptions in the main text.

We thank you for your time and hope the manuscript now meets the requirements of Communications Biology.

On behalf of all co-authors
Sincerely,

Ikki Matsuda & Hiroki Koda

See detailed responses to reviewers down below

Reviewer 2:

This manuscript provides interesting new data on the association between canine length, nose length, and body size in proboscis monkeys. The authors find a surprising negative correlation between body size and canine length in adult males and explain this by suggesting possible ecological costs for having long canines. This work will be of interest to others studying trade-offs in signaling in animals. For the most part the manuscript is well-written and the data are analyzed appropriately. The inclusion of simulations on the effect of costly canine development helps bolster the conclusions. I have just a few comments to improve the clarity and flow.

9	Our response	Changes made in the text (lines refer to the revised version)
The Introduction could be improved by citing the two papers by Bergman and Sheehan (2013 and 2016) on quality signaling and individual recognition in primates and discussing the evolution of badges of status in light of the potential multi-level society in proboscis monkeys. Also, Rohwer's (1982) important paper on badges of status should probably be cited (references provided below).	Thank you very much for providing us with key references. In the introduction section, we added the suggested references and briefly introduced the relationship between status badge and social cognition in relation to primate multilevel social system.	Addition made, L 52-57 Additionally, such badges of status may be related to animal social recognition⁵; status badges may be beneficial in large/complex societies like primate multilevel society, in which two or more levels of organization are recognizable. A status badge would therefore provide a useful solution for individual identification and for acquiring social recognition such as the relative dominance of conspecifics^{6,7}.
The sample size of males that was actually included in the study needs to be clarified. Lines 76-82 are not clear. Did Koda et al. study the same males as Stark et al.? How many females were included? I think a table detailing the study subjects would be helpful here.	We fully agree on your suggestion and have added the sample size. We also included a new table to show the detailed information of the study subjects (see the supplementary information: SI 1).	Addition made, L 91-93 All study subjects were adults, i.e., 18 harem-holding males and 10 females including one pregnant and two lactating individuals (see SI 1 for further detailed study subject information).
In the Results/Discussion, the paragraph on lines 138 to 142 needs further fleshing out and explanation. On lines 138-140, what is meant by the idea that "female nose size is linked to the effect on males"? Isn't it just that larger individuals have larger body parts and this includes noses? Or are the authors suggesting that positive selection on male nose size is driving larger noses in females? I'm not sure that these two possibilities can be teased out with this data set. It may be best to discuss both. On lines 140-142, what "activity" is being referred to? Canine activity or feeding etc.? This is not clear.	We meant the latter explanation, i.e., "positive selection on male nose size is driving larger noses in females". We, therefore, clarified this in the text. Additionally, we previously used the term "activity" to mean general primate daily activities including feeding, moving and social behaviours. We have clarified this point.	Changes made, L 157-162 Contrarily, whereas females do not compete through sound as do males, they unexpectedly exhibited the same nose size/body size correlation, suggesting that female nose size is linked to the effect on males; positive selection on larger male nose size may be driving larger noses in females. Female noses are probably sufficiently small to avoid interfering with their activity (e.g., feeding, moving, social behaviors, etc.), and thus, selection toward sex-differentiated expression does not occur.
Finally, the passive voice is used often throughout the manuscript. It would benefit from a switch to the active voice.	We improved throughout the text.	Please see the text.
L 22-25 – The first sentence is very long. Breaking it up would make it clearer.	We improved.	Change made, L 23 The uniquely enlarged noses of male proboscis monkeys (Nasalis larvatus) are prominent adornments, and a sexually selected male masculine trait. A recent study showed the significant correlations among nose, body, and testis sizes and the clear association between nose size and the number of females in a male's harem.
L 28 – Add "A" before "similar".	We improved.	Changes made, L 28 A similar relationship between nose and body size was observed in females, whereas only a weak correlation was noted between canine and body size.
L 33 – I don't think "However" belongs here, since you are presenting an additional reason why longer canines are	We improved.	Changes made, L 32 Additionally, longer canines are opposed by natural selection because

disadvantageous. "In addition," or something similar would be better.		they impose a larger gape upon its bearer and reduce foraging efficiency, particularly in folivores.
L 48 – This is does not seem right, "ornaments independently evolve for a function in combat." Do you mean "ornaments independently evolve FROM a function in combat and instead act as a partner attraction signal"?	Thank you. We improved.	Changes made, L 48 By contrast, ornaments independently evolve from a function in combat and instead act as a partner attraction signal (ultimately determined by female choice).
L 83 – Start a new paragraph with the line "A linear model".	We improved.	Changes made, L 98
L 111 – Since it is likely body mass driving larger noses for females, I think the wording here would be better as, "was positively associated with body mass".	We improved.	Changes made, L 129-130 Conversely, for females, the best-fitting model only included nose size, which was positively associated with body mass.
L 134 – Change "vocalization opposed" to "vocalizations as opposed".	We improved.	Changes made, L 153 Thus, a larger body, which contributes to male athleticism, would allow more effective displays of fighting ability, whereas nose size is the best indicator of fighting ability because it is easier to assess via vocalization as opposed to visual examination.
I hope these comments are helpful in revising your work.	Thank you very much for your comments. It was really helpful to improve our manuscript.	None

Reviewer 3:

This manuscript reported a negative relationship between canine size and body size in morphologically unique proboscis monkeys. A negative relationship is unexpected from the sexual selection theory but can be understood based on a unique characteristic of proboscis monkeys. Morphological data in wild proboscis monkeys is valuable. The manuscript is very simple and easy to understand. But I have several questions and comments.

THANK YOU VERY MUCH FOR YOUR VALUABLE COMMENTS. WE APPRECIATED ALL YOUR POSITIVE COMMENTS.

Reviewer's comment (lines refer to the original submission)	Our response	Changes made in the text (lines refer to the revised version)
First, I do not think that a section of simulation is necessary. It adds little to the results.	I undersand the point suggested by the reviewer, but in our view, clarifying our observation-based hypothesis (costly canine hypothesis) by the mathematical simulation would be important but a bit challenging. Therefore, it would be great to allow us to keep this simulation results; please also note that the editor is suggesting to keep this simulation result in the decision letter.	None
Second, it would help if authors explain details of proximate aspects (mechanism) of canine development. In 62-64, the authors mention that canines develop in response to testosterone. But no citation on these statements. Is this true?	The effects of testosterone on sexually dimorphic musculature are well studied (Bhasin et al. 2012; Muller 2017), but the reviewer is correct that few studies have investigated the relationship between testosterone and canine growth (but see Zingesser and Phoenix 1978). As suggested by the reviewer, we added some references in the text. On the other hand, we did not add the detailed explanation for those of developping mechanism as we do not incorporate such hormone data in our study. Bhasin S, Jasuja R, Serra C, Singh R, Storer TW, Guo W, Travison TG, Basaria S, Nieschlag E, Behre HM, Nieschlag S (2012) Androgen effects on the skeletal muscle. In: Nieschlag E, Behre HM (eds) Testosterone: Action, Deficiency, Substitution. Cambridge University Press, Cambridge, pp 191-206 Muller MN (2017) Testosterone and reproductive effort in male primates. Horm Behav 91:36-51. Zingesser MR, Phoenix CH (1978) Metric characteristics of the canine dental complex in prenatally androgenized female rhesus monkeys (Macaca mulatta). Am J Phys Anthropol 49:187-192.	Addition made, L 69 Sexually dimorphic male traits (e.g., body mass, nose, testis, and canines) develop after sexual maturation, primarily in response to endocrine changes in the levels of androgens such as testosterone ^{10, 11, 12} .
L76-77: 18 free-ranging adult male proboscis monkeys. Are those males harem-holders? Or a mixture of harem-holders and bachelors? This information should be important because authors assume that differences between harem-holders and others are important.	We clarified this point in the text.	Addition made, L 91-93 All study subjects were adults, i.e., 18 harem-holding males and 10 females including one pregnant and two lactating individuals (see SI 1 for further detailed study subject information).
L83-87: Although what is written is	We had described the response and	Changes made, L 98-99

correct, I appreciate it if authors could explain which one is a response variable and what are explanatory variables more clearly.	explanatory variables in the text as follows: “To obtain a linear dimension comparable to the canine dimension, body mass and nose size were cube root- and square root-transformed, respectively, to generate response and explanatory variables“. However, we agree that it might not be so clear in the text, so this text has been clarified.	A linear model was used to establish whether body mass (response variable) was related to other physical properties such as nose and canine size (explanatory variables).
L129-132. Although I am not a native English speaker, I prefer not to use two which in a single sentence.	Thank you for your suggestion. We improved the text.	Changes made, L 149 Additionally, among proboscis monkey males, nose size, which is highly correlated with body size, determines the formant frequencies of loud vocalizations, that may decide contests without any direct physical competition, or at least direct biting^{5, 7, 16}.
Table 1. please use the same number of decimal digits for all values.	We improved.	See the Table 1

Reviewer 4:

This is an interesting paper on the relationship between canine tooth size, body size, and nose size in proboscis monkeys. The authors have some valuable data, and some interesting observations that might have some interesting implications for the evolution of proboscis monkey noses. However, the paper suffers from an inadequate explanation of the materials and methods, inadequate presentation of the results, and failure to consider alternative explanations for the observed results.

THANK YOU VERY MUCH FOR YOUR FRUITFUL COMMENTS. WE DID OUR BEST TO IMPROVE THE TEXT BASED ON YOUR COMMENTS.

Reviewer's comment (lines refer to the original submission)	Our response	Changes made in the text (lines refer to the revised version)
The introduction is mostly fine, but there are a few confusing statements. For example, the authors state “By contrast, ornaments independently evolve for a function in combat, reflecting their function as a partner attraction signal (ultimately determined by female choice).” This doesn’t make sense. There are few more rough spots in the introduction. I think that the authors need to carefully review exactly what they are saying and make sure that the introduction is both concise and clear.	This part suggested by the reviewer was also criticized by the other reviewer. We carefully improved the text.	Changes made, L 47-49 By contrast, ornaments independently evolve from a function in combat and instead act as a partner attraction signal (ultimately determined by female choice).
Methods. The data are referred to another paper, but the reader should not be forced to chase the data down. What exactly is nose length and breadth? How was body mass measured? Were pregnant female included? Did you note lactation?	We added the detailed information in the methods section.	Addition made, L 80-97 Between July 2011 and December 2016, we captured 28 free-ranging adult proboscis monkeys in the Lower Kinabatangan Floodplain, Sabah, Borneo, Malaysia (5°18’N to 5°42’N and 117°54’E to 18°33’E). To reduce the impact of capturing on the animal’s social system, we captured all study subjects during the night (see detailed methods and ethical statement of Stark et al.¹⁵). While animals were anaesthetized, we performed in-situ measurements for their body parts using a scale (body mass) and caliper (nose size and canine length). In this study, we additionally included data for maxillary canine length to those on body mass and nose size (length × width) for the same 18 males obtained by Koda et al.⁹ as well as data for 10 female canine length, body mass, and nose size. Noted that canine length refers to the apex to base measurement (the height of the crown), and the maxillary canine length was measured because of its larger size and greater importance in behavioral displays and weaponry¹⁶. All study subjects were adults, i.e., 18 harem-holding males and 10 females including one pregnant and two lactating individuals (see SI 1 for further detailed study subject information). The veterinarian in the sampling team attempted in-situ age/class estimation for the study subjects based on the three wear level of their molars, i.e., low: young adult; medium: adult; high: old adult, although the estimation was rather rough due the difficult in-situ condition in the forest at

When measuring canines, did you account for wear? Or did you need to? Canine length can refer to the apex to base measurement (the height of the crown), which has been argued to be the functional aspect of the canine, or the mesiodistal length at the base of the tooth. This makes a huge difference, because if you are measuring canine tooth height, wear will be expected to shorten the tooth with age, and if size increases with age, then perhaps your results are an artifact of age-size and tooth wear. On the other hand, if you are measuring the occlusal dimension of the tooth, then you might run into problems because this aspect of the tooth is not as strongly correlated with behavioral measures as height.	Thank you very much for your comments to clarify our methodology. As you expected, canine length can refer to the apex to base measurement (the height of the crown). On the other hand, a rough age/class estimation was conducted using the wear level of the molars but not canines. However, because we captured the animals in the forest at night and performed all measurements there in the forest, our estimation would not be so accurate due to the such in-situ condition. We added the estimated age/class information for each individual in the supplementary information (SI1) and partly showed in Figure 1.	night, e.g., limited tools, light and time the animal remained anaesthetized. Changes made, L 88-91 Noted that canine length refers to the apex to base measurement (the height of the crown), and.... Also see the Figure 1 and SI 1 (Supplementary information 1) for the addition of age/class information.
I am not an expert on the mathematical modelling, and so cannot really comment on it. Models are great, but better explanation is needed. After all, the test of the hypothesis should be based on the data. What bothers me here is that I have some fair experience in stats, and when I read a paper I should know exactly what you are doing, and I cannot see that here. How is the termination of canine tooth development determined? These are wild data, and I assume that sample sizes (which I cant determine from this study – all I know is that data for 18 males and an unknown number of females was added to some data from another study that I don't have) are small. In the figure I count 10 females and 18 males, which for the males matches the stated number gathered, but if these were added to other data then something is wrong.	First, we worry that the reviewer may have misunderstood what we did through the mathematical simulation. Our mathematical simulation is different from the one like statistical modeling using the real/observed data set. But, the mathematical simulation in this manuscript was to support our proposed hypothesis (costly canine hypothesis) based on the results revealed by the real/observed data set. To clarify, what we did in this manuscript was the following:  1) correlation test using the real/observed data set via regression (linear model) analyses and ANCOVA 2) based on the above analyses, we proposed the “costly canine hypothesis” 3) to confirm our hypothesis, we performed the mathematical simulation 4) our hypothesis was supported through the mathematical simulation Second, as the reviewer suggested, current description for the mathematical simulation part is very minimal, and thus it might be difficult to fully understand our simulation part only from the main text. However, please note that we uploaded the detailed explanation with some figures to the preprint server as the supplementary information. We had decided not including the detailed explanation in the main text because to smoothly and efficiently read/understand the text/findings, it interrupted by those detailed descriptions with many formulas. Lastly, as described in the supplementary information, the termination of canine	None

	tooth development was referring to previously published data (not from our original data) like Cercopithecus (Leigh et al. 2005) and proboscis monkeys (Schultz 1942). Leigh SR, Setchell JM, Buchanan LS. Ontogenetic bases of canine dimorphism in anthropoid primates. Am J Phys Anthropol 127, 296-311 (2005). Schultz AH. Growth and development of the proboscis monkey. Bulletin of the Museum of Comparative Zoology 89, 277-314 (1942).	
Reporting of the results needs to be much more systematic. The authors have done regressions, ANCOVA, and something else, but I can't be quite sure because the results are not actually here. What I do have is some graphs, and the graphs are suggesting several things. First, nose size and body size are tightly correlated in females, an less so in males, and it looks like a standard allometric phenomenon with an offset between the sexes (the ANCOVA should show no significant interaction and the slope between males and females does not significantly differ). The canines show no relation between female canine size and body size, and the males show a negative relationship, though it looks like the correlation is going to be weak. These are interesting results, but I am not convinced that the authors have considered alternative explanations. First, if male canine size is impacted by wear, but size increases with age (see for example drills), then we should expect canine size to have a negative correlation with body size not through selection, but through tooth wear. Second, male nose size increases with size. If size increases with age, then the model proposed by the authors is not supported, as the results reflect an interaction between age, size, and tooth wear. Alternatively, if male canine tooth eruption is completed before final adult body mass is achieved, then we should expect a loose, negative correlation between tooth size and body size. Again, this would mean that the model of the authors is not supported. I really can't say much more about the results and discussion unless this basic issue is addressed. The authors need to address this issue head on by including age in their estimates. Without that, not much can be said. The models might	We agree on some points suggested by the reviewer. The age/wear effects on our results would still be unclear, but we newly added the age/class information for each individual, which was roughly estimated by the wear level for the molars. As described in the methods section, the estimation was not very accurate and the sample size was very small, but the values for the young/old adult males were still in a range of the ones for "averaged" adult males. We, therefore, believe that the proposed effects of age/wear by the reviewer on the results of our analysis may be less, though we still need to carefully confirm this using larger data set in future. And so, we added the limitation for the analysis in the main text. We also agreed the point suggested by the reviewer, i.e., it is necessary to clarify the developmental mechanism for the canine, body mass and nose to further understand our proposed hypothesis. However, we would also need to note that we selected ONLY adult individuals (we selected physically mature individuals) in our analysis, which would be able to minimize the variation in body features depending on their different growing stages (age). Thus, we believe that the effects of age/wear on the current results would be minimized. As we can not re-measure the body parts from the same animals and can not add new data from free-ranging proboscis monkeys in the wild, we hope our addition made will be acceptable for the reviewer.	Addition made, L 125-129 As the values for the young/old adult males were within the range of the ones for the "averaged" adult males (Fig. 1), we, therefore, believe that the effects of age/wear of each individual male on this statistical tendency would be less. However, due to the limited sample size, further analysis using larger data set would be necessary to reconfirm these results in a future study. See the figure 1 (newly added the information about the age/class estimation for all study subjects).

work under the assumptions that authors input, but I am not sure that the modelling accounts for these two basic features. If it does, the authors need to explain their model better, and it needs to be validated against actual data ("ground tested").

--

--

Reviewers' comments:

Reviewer #2 (Remarks to the Author):

The authors have done a good job of responding to all of the comments by the reviewers. The manuscript will be an excellent addition to the literature. I just have a few small changes that need to be added.

L 30 – Was the correlation in females also negative?

L 55 – Change “society” to “societies”.

L 56 – Change to “useful solution in lieu of individual identification for acquiring an understanding of the relative...”.

L 90 – Change “female” to “female’s”.

L 91- Change “Noted” to “Note”.

L 97 – Add “categories of” after “three”.

L 101 – What kind of linear model? Regression? Mixed-effects?

L 131 – Add “a” before “larger”.

L 170 - Consider changing the first “effectiveness” to “efficacy” to avoid reusing the word multiple times.

L 179 – Again, note here whether the correlation for females was positive or negative.

Reviewer #3 (Remarks to the Author):

I thank the authors for responding to my previous comments. I understand that authors are forced to shorten their explanations because of the length limitation. Based on this in my mind, I feel that the simulation and Fig 2 should not be in the main text more strongly. Readers never understand the details of the simulation and what Fig 2 means by reading the main text. I think that it is better to remove Fig 2 from the main text (and accordingly L38-39 in Abstract; L114-122 in Methods) and to mention that authors did simulation very briefly in Discussion.

COMMSBIO-19-1713A

Reply to Reviewers

Dear reviewers,

Thank you very much for your valuable comments and suggestions. Please find attached the revised manuscript with all revised text in grey shading. Our response to specific reviewer comments are given at the bottom of this letter.

We thank you for your time and cooperation.

Sincerely,

Ikki Matsuda & Hiroki Koda

See detailed responses to reviewers down below

Reviewer 2:

The authors have done a good job of responding to all of the comments by the reviewers. The manuscript will be an excellent addition to the literature. I just have a few small changes that need to be added.

THANK YOU VERY MUCH FOR YOUR VALUABLE COMMENTS. WE IMPROVED THE TEXT BASED ON YOUR SUGGETIONS.

Reviewer's comment (lines refer to the original submission)	Our response	Changes made in the text (lines refer to the revised version)
L 30 – Was the correlation in females also negative?	It was a very weak negative effect, but the $\Delta AICc$ was over 2.0, and thus it would not be necessary to consider such effect. We clarified the effect as suggested by the reviewer.	Additon made, L 29: A similar relationship between nose and body size was observed in females, whereas only a weak negative correlation was noted between canine and body size.
L 55 – Change “society” to “societies”.	This suggestion has been taken.	Changes made, L 54: Additionally, such badges of status may be related to animal social recognition ⁵ ; status badges may be beneficial in large/complex societies like primate multilevel societies , in which two or more levels of organization are recognizable.
L 56 – Change to “useful solution in leu of individual identification for acquiring an understanding of the relative...”.	This suggestion has been taken.	Changes made, L 56-57: A status badge would therefore provide a useful solution in lieu of individual identification for acquiring an understanding of the relative dominance of conspecifics ^{6, 7} .
L 90 – Change “female” to “female’s”.	This suggestion has been taken.	Changes made, L 91: In this study, we additionally included data for maxillary canine length to those on body mass and nose size (length × width) for the same 18 males obtained by Koda et al. ⁹ as well as data for 10 females’ canine length, body mass, and nose size.
L 91- Change “Noted” to “Note”.	This suggestion has been taken.	Changes made, L 91: Note that canine length refers to the apex to base measurement (the height of the crown), and the maxillary canine length was measured because of its larger size and greater importance in behavioral displays and weaponry ¹⁶ .
L 97 – Add “categories of” after “three”.	This suggestion has been taken.	Additions made, L 99: The veterinarian in the sampling team attempted in-situ age/class estimation for the study subjects based on the three categories of wear level of their molars, i.e., low:....
L 101 – What kind of linear model? Regression? Mixed-effects?	We used a linear regression mode, so that we clarified it in the text.	Additions made, L 102: We used a linear regression model to establish whether body mass (response variable) was related to other physical properties such as nose and canine size (explanatory variables).
L 131 – Add “a” before “larger”.	This suggestion has been taken.	Additions made, L 132: However, due to the limited sample size, further analysis using a larger data set would be necessary to reconfirm these results in a future study.
L 170 - Consider changing the first “effectiveness” to “efficacy” to avoid reusing the word multiple times.	This suggestion has been taken.	Changes made, L 171: The prominent nose may interfere with effective canine bites, thereby reducing their efficacy as weapons.

L 179 – Again, note here whether the correlation for females was positive or negative.	Please see the above comments.	Additions made, L 180: Contrary to the significant negative slope for the male canine size-body size correlation, we only found a weak negative correlation for the female slope.
---	---------------------------------------	---

Reviewer 3:

I thank the authors for responding to my previous comments. I understand that authors are forced to shorten their explanations because of the length limitation. Based on this in my mind, I feel that the simulation and Fig 2 should not be in the main text more strongly. Readers never understand the details of the simulation and what Fig 2 means by reading the main text. I think that it is better to remove Fig 2 from the main text (and accordingly L38-39 in Abstract; L114-122 in Methods) and to mention that authors did simulation very briefly in Discussion.

THANK YOU VERY MUCH FOR YOUR COMMENTS. ALTHOUGH THE REVIEWER RECOMMENDED US TO MINIMIZE THE SIMULATION PART AND DELETE THE FIGURE 2, WE DECIDED TO KEEP THE DESCRIPTION OF THE SIUMULATION IN THE MAIN TEXT WITH MAKING SOME ADDITIONS TO EXPLAIN MORE ABOUT OUR MATHEMATICAL SIMULATION (LINES 192-201). ADDITIONALL, THE README FILE IN THE GITHUB LINK WAS MOVED TO THE MAIN METHODS AS SUPPLIMNATARY INFOMRAION (SI 2). WE DECIDED TO DO THIS AFTER THE DISCUSSION WITH THE HANDLING EDITOR, AND SO WE REALLY HOPE THAT YOU UNDERSTAND THIS.

REVIEWERS' COMMENTS:

Reviewer #5 (Remarks to the Author):

I am reviewing this manuscript for the first time and was specifically asked to comment on the soundness and interpretation of the mathematical model. I have consequently refrained from commenting on the statistics etc., even though in some places I was less than convinced by the authors' approach. I checked the mathematical model in detail and found no errors. In essence, the model shows that if long canines reduce feeding efficiency, then this can generate a negative correlation between canine size and adult body size. In my view this is an obvious point that does not necessarily require a mathematical model. However, the authors' attempt to fit the model to data adds some interesting realism that partially justifies the rather complex model.

I found the authors' discussion of the role their model plays in this study a bit confused. They write:

'Concerning the mechanism underlying the trade-off among body mass, nose size, and canine size, we predict that canine growth would actually exhibit a negative correlation with testosterone levels, opposite to the common and ancestral situation. To test this "costly canine hypothesis", we conducted preliminary mathematical simulations considering body mass development with a growth constrained by canine size.'

The model does not even include testosterone, so it couldn't possibly test this hypothesis. In any case, models can only test the logic of an argument, and not an empirical hypothesis. I think it's more accurate to say that the model shows that a negative correlation between body and canine size could arise if long canines reduce feeding efficiency. The predictions about the role of sex and male status could also be described in simple terms. Any grander claims are unwarranted.

A couple of other comments:

1. The default expectation for both males and females would be that nose and canine size are proportional to body size. As such, a positive correlation between nose size and body size in either sex is not surprising in itself and requires no complex explanation. For this reason, relative sizes of ornaments and weapons are often used. In many species with sexually selected ornaments, the ornament has positive allometric slope. I found the authors discussion of this issue a bit lacking in rigor.

2. Unless I missed it, the authors do not mention whether females fight using their canines. This is important point for interpreting the results.

And a few minor points:

Line 45: 'many male-specific characters' would be better, to avoid the impression that there are no other evolutionary explanations for such characters

Line 46: 'are' would be better than 'represent' (also line 58)

Line 48: 'ornaments independently evolve from a function in combat': This reads quite obscurely. In any case some ornaments might have originally evolved for combat but subsequently lost this function. Maybe something simpler like 'are not involved in combat'?

Line 76: I'm not sure what 'suggestive' means here

Line 79: add 'in interspecific comparisons' or similar

Line 114: Here you talk about 'the model simulation' as if it had already been introduced, but it hasn't. Perhaps change to something like 'We also developed simulation models with the aim to...'

Line 127: 'although the model...': This clause would make more sense at the end of the previous sentence.

Line 128-131: This sentence is quite hard to understand. I think you mean that age and wear do not contribute greatly to the relationship between body size and canine size. Please rephrase.

Line 155: 'best indicator': Best in what sense? Suggest rephrasing.

COMMSBIO-19-1713B
Reply to Reviewer

Dear reviewer,

Thank you very much for your valuable comments and suggestions. Please find attached the revised manuscript with all revised text in **grey shading**. Our response to specific reviewer comments are given at the bottom of this letter.

We thank you for your time and cooperation.

On behalf of all co-authors
Sincerely,

Ikki Matsuda & Hiroki Koda

See detailed responses to reviewers down below

Reviewer 5:

I am reviewing this manuscript for the first time and was specifically asked to comment on the soundness and interpretation of the mathematical model. I have consequently refrained from commenting on the statistics etc., even though in some places I was less than convinced by the authors' approach. I checked the mathematical model in detail and found no errors. In essence, the model shows that if long canines reduce feeding efficiency, then this can generate a negative correlation between canine size and adult body size. In my view this is an obvious point that does not necessarily require a mathematical model. However, the authors' attempt to fit the model to data adds some interesting realism that partially justifies the rather complex model.

THANK YOU VERY MUCH FOR YOUR TIME TO VARIFY OUR SIMULATION SECTION WITH SOME VALUABLE COMMENTS. WE DID OUR BEST TO IMPORVE OUR MANUSCRIPT BASED ON YOUR COMMENTS (SEE BELOW).

Reviewer's comment (lines refer to the original submission)	Our response	Changes made in the text (lines refer to the revised version)
I found the authors' discussion of the role their model plays in this study a bit confused. They write: ‘Concerning the mechanism underlying the trade-off among body mass, nose size, and canine size, we predict that canine growth would actually exhibit a negative correlation with testosterone levels, opposite to the common and ancestral situation. To test this “costly canine hypothesis”, we conducted preliminary mathematical simulations considering body mass development with a growth constrained by canine size.’ The model does not even include testosterone, so it couldn't possibly test this hypothesis. In any case, models can only test the logic of an argument, and not an empirical hypothesis. I think it's more accurate to say that the model shows that a negative correlation between body and canine size could arise if long canines reduce feeding efficiency. The predictions about the role of sex and male status could also be described in simple terms. Any grander claims are unwarranted.	We fully agreed the comments by the reviewer, and so we toned down all parts where we discussed the results of our mathematical simulation. Additionally, we clearly stated that we did not test the factor about testosterone in the text. On the other hand, concerning the comments “predictions about the role of sex and male status could also be described in simple terms”, it may be possible to simplify this parts, though we decided retaining the previous explanation about the methods and results for mathematical simulation section (even though a bit wordy) as readers may range from backgrounds as theoretical biologist to more wide ranging biological fields, and thus we believe that such careful explanations would be necessary.	Changes made, L 341-354: Concerning the mechanism underlying the trade-off among body mass, nose size, and canine size, we predict that canine growth would actually exhibit a negative correlation possibly with testosterone levels, opposite to the common and ancestral situation³³. To test a part of this “costly canine hypothesis”, we conducted preliminary mathematical simulations considering body mass development with a growth constrained by canine size. Echoing our empirical findings, the negative relationship between body and canine size was observed only in males when we assumed a large cost of canine development (Fig. 3); noting that the model did not consider the possible basal factors such as testosterone level which might fundamentally determine the maturation patterns of the sexual traits, due to no available data for the initial parameter proposals. Our study, however, raises interesting questions regarding the developmental mechanism that produces a negative correlation among adult body, nose size, and canine size. Future detailed studies focusing on the behavior and development system of proboscis monkeys are therefore required.
The default expectation for both males and females would be that nose and canine size are proportional to body size. As such, a positive correlation between nose size and body size in either sex is not surprising in itself and requires no complex explanation. For this reason, relative sizes of ornaments and weapons are often used. In many species with sexually selected ornaments, the ornament has positive allometric slope. I found the authors discussion of this issue a bit lacking in rigor.	We realized that the suggested point is really important, but we did not mention it in our previous version. We included this point in the discussion section in this version. Thank you very much!	Additions made, L 321-316: In other respects, it has been found that secondary sexual characteristics, i.e., ornaments or weapons, have a positive allometric slope concerning body size in a number of animal species^{23, 34}, a positive correlation between nose size and body size in either sex would just be a natural phenomenon and requires no complex such explanation.
Unless I missed it, the authors do not mention whether females fight using their canines. This is important point for interpreting the results.	We also had left out this important point in the previous version. We included the description about female agonistic behaviours in the text.	Additions made, L 329-333 Contrary to the significant negative slope for the male canine size-body size correlation, we only found a weak negative correlation for the female slope. Considering lower incidence of agonistic interaction, either using

		canines and leaping between trees, with weak or absence of linearity in hierarchy among females within groups ^{13,38} , such canine size-body size correlation may be less prominent in females than that in males.
Line 45: 'many male-specific characters' would be better, to avoid the impression that there are no other evolutionary explanations for such characters	This suggestion has been taken.	Changes made, L 37: Darwinian evolutionary theory explains many male-specific characteristics as a consequence of sexual selection ¹
Line 46: 'are' would be better than 'represent' (also line 58)	This suggestion has been taken.	Changes made, L 39, 50: Large canines in some mammals, including primates, are a typical example of a weapon used in contests with conspecific male rivals ² . Proboscis monkeys (Nasalis larvatus) are a typical sexually dimorphic primate with male-specific enlarged noses that are prominent adornments, which have been linked with their sexually competitive social system of one-male groups, suggestive of a multilevel social system ⁸ .
Line 48: 'ornaments independently evolve from a function in combat': This reads quite obscurely. In any case some ornaments might have originally evolved for combat but subsequently lost this function. Maybe something simpler like 'are not involved in combat'?	This suggestion has been taken.	Changes made, L 40: By contrast, ornaments are not involved in combat and instead act as a partner attraction signal (ultimately determined by female choice).
Line 76: I'm not sure what 'suggestive' means here	We deleted "suggestive" in the text.	Deleted
Line 79: add 'in interspecific comparisons' or similar	This suggestion has been taken.	Additions made, L 67-70: Therefore, male canines are expected to serve as important weapons for mate competition or defense, especially because canine size generally reflects the strength of intra-sexual selection more directly than body size in interspecific comparisons ^{2,14} .
Line 114: Here you talk about 'the model simulation' as if it had already been introduced, but it hasn't. Perhaps change to something like 'We also developed simulation models with the aim to...'	This section has largely been re-written with additions of a lot of methodological explanations about our mathematical simulations. Please see in the text, lines xx-xx.	See in the text, L 126-241.
Line 127: 'although the model...': This clause would make more sentence at the end of the previous sentence.	This suggestion has been taken.	Changes made, L 257-258: The best-fitting model included both male nose and canine size, although the fit of the model including only canine size was almost as good (i.e., $\Delta AICc < 2.0$; Table 1).
Line 128-131: This sentence is quite hard to understand. I think you mean that age and wear do not contribute greatly to the relationship between body size and canine size. Please rephrase.	This suggestion has been taken.	Additions made, L 260-263: As the values for the young/old adult males were within the range of the ones for the "averaged" adult males (Fig. 1), we, therefore, believe that age/wear of each individual male would not contribute greatly to the relationship between body size and canine size.
Line 155: 'best indicator': Best in what sense? Suggest rephrasing.	We clarified the point.	Changes made, L 300-303: Thus, a larger body, which contributes to male athleticism, would allow more effective displays of fighting ability, whereas nose size would be a more suitable proxy to represent fighting

		ability because it is easier to assess via vocalization as opposed to visual examination.
--	--	---